# Stress Distribution in Wear Analysis of Nano-Y_2_O_3_ Dispersion Strengthened Ni-Based μm-WC Composite Material Laser Coating

**DOI:** 10.3390/ma17010121

**Published:** 2023-12-26

**Authors:** Li Tao, Yang Yang, Wenliang Zhu, Jian Sun, Jiale Wu, Hao Xu, Lu Yan, Anhui Yang, Zhilong Xu

**Affiliations:** 1Department of Robot Engineering, School of Mechanical Engineering, Jiangsu Ocean University, Lianyungang 222005, China; taoli0401@163.com (L.T.); yangyang5201022@163.com (Y.Y.); zhuwenliang@mail.chinamso.com (W.Z.); 2023210507@jou.edu.cn (J.W.); 2021121053@jou.edu.cn (H.X.); 2021121055@jou.edu.cn (L.Y.); 2021121056@jou.edu.cn (A.Y.); 2021121054@jou.edu.cn (Z.X.); 2School of Mechanical and Electrical Engineering, Xi’an Polytechnic University, Xi’an 710048, China

**Keywords:** ODS, Ni-based alloy coating, nano-Y_2_O_3_, μm-WC, dispersion strengthen

## Abstract

Oxide-dispersion- and hard-particle-strengthened (ODS) laser-cladded single-layer multi-tracks with a Ni-based alloy composition with 20 wt.% μm-WC particles and 1.2 wt.% nano-Y_2_O_3_ addition were produced on ultra-high-strength steel in this study. The investigation of the composite coating designed in this study focused on the reciprocating friction and wear workpiece surface under heavy load conditions. The coating specimens were divided into four groups: (i) Ni-based alloy, nano-Y_2_O_3_, and 2 μm-WC (2 μm WC-Y/Ni); (ii) Ni-based alloy with added 2 μm-WC (2 μmWC/Ni); (iii) Ni-based alloy with added 80 μm-WC (80 μmWC/Ni); and (iv) base metal ultra-high-strength alloy steel 30CrMnSiNi2A. Four conclusions were reached: (1) Nano-Y_2_O_3_ could effectively inhibit the dissolution of 2 μm-WC. (2) It can be seen from the semi-space dimensionless simulation results that the von Mises stress distribution of the metal laser composite coating prepared with a 2 μm-WC particle additive was very uniform and it had better resistance to normal impact and tangential loads than the laser coating prepared with the 80 μm-WC particle additive. (3) The inherent WC initial crack and dense stress concentration in the 80 μm-WC laser coating could easily cause dislocations to accumulate, as shown both quantitatively and qualitatively, resulting in the formation of micro-crack nucleation. After the end of the running-in phase, the COF of the 2 μm-WC-Y_2_O_3_/Ni component samples stabilized at the minimum of the COF of the four samples. The numerical order of the four COF curves was stable from small to large as follows: 2 μm-WC-Y_2_O_3_/Ni, 2 μm-WC/Ni, 80 μm-WC/Ni, and 30CrMnSiNi2A. (4) The frictional volume loss rate of 2 μm-WC-Y_2_O_3_/Ni was 1.3, which was significantly lower than the corresponding values of the other three components: 2.4, 3.5, and 13.

## 1. Introduction

Oxide-dispersion-strengthened (ODS) alloys are used for the surface strengthening of typical high-speed, heavy-duty, and especially wear-resistant workpieces, such as an aero engine spindle powertrain. Extremely uniform nano-Zr(Y)O_2_ dispersion-strengthened W alloys can be prepared using the hydrothermal method combined with hot isostatic pressing [1]. Nano-Y_2_O_3_ addition in Ni-based alloys can provide microstructural refinement. This is because Y_2_O_3_ has good thermal stability and oxidation stability at high temperatures and is able to maintain its nano size and dispersion state while forming strong chemical reactions with Ni and other alloying elements in Ni-based alloys, thus forming strong oxide particles at the grain boundaries. Nickel- and iron-based ODS superalloys can be used in durable TPS/thermal structures for hypersonic vehicles, such as PM ferrite- and nickel-based oxide yttrium dispersion-strengthened alloys as honeycomb panels and fasteners for PM 1000/2000 [2]. By optimizing the precursor acidity [3] and reducing the initial particle size [4], the ODS doping process of tungsten alloy with high-performance oxide dispersion can be prepared. The addition of nano-CeO_2_ can improve the solid solubility of Cr in the nickel-based superalloy, the microhardness in the microstructure can be effectively homogenized, and the resistance of the composite under a thermal shock environment can be enhanced [5]. Both La_2_O_3_ and Y_2_O_3_ in MoSi_2_ generate new insoluble compounds, which are dispersed inside the grains and at the grain boundaries, and have a fine microstructure, and thus, the strength and toughness of molybdenum can be significantly improved [6]. The addition of submicron-WC-reinforced particles in an iron-based alloy can induce plasticity via phase transformation, grain refinement, non-equilibrium grain boundary strengthening, and the nanoscale precipitation effect, and comprehensively improve the compressive strength, fracture strain, ultimate tensile strength, and elongation [7]. Directional recrystallization of nickel-based superalloys can be achieved by 3D laser printing to selectively enhance the fatigue or creep properties of large columnar grains [8]. Dispersion strengthening and fine grain strengthening can improve the corrosion resistance, radiation resistance, and high-temperature creep resistance of nuclear cladding materials. The formation and composition design parameters of typical oxides Y-Ti-O, Y-Al-O, and Y-Zr-O in oxide-dispersion-strengthened (ODS) FeCrAl alloys have a great influence on the properties of nuclear-industry-grade ODS FeCrAl alloys [9]. The perfluorosulfonic acid membrane composite of rare earth elements europium (III) and terbium (III) chlorides has a maximum dispersion and corresponding loss [10]. Deep machine learning was used to determine the strong coupling relationship between the relative surface hardness of electroless nickel plating on a laser-formed workpiece and the scratch width parameter [11]. The optimization method of annealing parameters via nanometer angle analysis for electroless nickel plating was used in [12]. Intermetallic compounds containing Nd and Pr show excellent wettability, and the presence of Ni, Cu, Fe, Nd, and Pr enhances WC densification [13]. Double perovskite oxides based on RE are suitable for thermoelectricity [14]. The thermal shock resistance and corrosion resistance of thermal spray coating are significantly improved after the addition of rare earth elements [15]. Some scholars believe that there are few studies on rare earth oxide laser cladding coatings on nickel-based superalloys, and all members of the research team where the author is working have reservations about this view [16]. WC—10% Co—4% Cr can significantly improve the friction and wear properties of a steel matrix [17]. The addition of nano-WC particles and La_2_O_3_ to La in the Ni-mass percentage 60%WC via a laser process correspondingly enhances the uniformity of the coating, the refinement of the microstructure, and the microhardness of the laser coating [18]. A content of 20% rare earth Er can be stabilized in high-entropy transition metal borides [19]. The 10 wt.% Cr + 2 wt.% high chromium and multi-walled carbon nanotubes WC-CoCr coatings show the highest wear resistance [20]. Vacancy plays an important role in improving the mechanical properties of thermoelectric materials containing La, Pr, and Nd [21]. The research and development of rare earth tungstate oxide nanostructures is of great importance [22]. The cooling rate of the laser molten pool is very important for WC decomposition [23]. Rare earth elements play an important role in reducing the dilution and crack sensitivity of a laser cladding layer and can effectively refine the microstructure [24]. The cobalt content of WC-CO carbide, the average grain size of WC, and the uniformity of microstructure affect the performance [25]. The addition of rare earth oxides can refine the microstructure of the coating, reduce the secondary dendrite spacing, and significantly improve the corrosion resistance [26]. Different ceramic particle sizes and volume fractions can improve the wear performance of aluminum alloy to different degrees [27]. The friction time series of sliding friction is self-similar, and its universal power spectrum is 1/f [28].

Although the coating prepared using the laser cladding process has a metallurgical bonding interface with a better bonding strength of the base metal, the coatings made using other processes, including cold spraying, thermal spraying, and air plasma spraying, also show vigorous vitality and high commercial and academic values in their respective applications.

Viorel Goanță’s team studied a WIP-C1 coating prepared using cold spraying on the surface of 4340 low alloy steel. The crack nucleation position and crack propagation trend of the coating under the effect of gravity in terms of the interfacial binding strength and alternating symmetric cyclic fatigue load produced the following results: the interface bonding strength between the two materials was satisfactory—all initial crack growth occurred on the substrate rather than at the interface or inside the coating, and the coating material only had an inhibitory effect on the surface damage of the substrate [29]. Michael P. Milz and his colleagues investigated the mechanical load fatigue behavior and salt spray corrosion resistance of a two-wire arc thermal-sprayed ZnAl coating on the surface of unalloyed steel for marine environment structural components (S355 JRC+C). By increasing the hardness and reducing the roughness and uniform coating thickness, shot peening could improve the high cycle fatigue resistance and corrosion fatigue resistance and prolong the service life of the ZnAl coating [30]. Satyapal Mahade et al. studied a high-durability thermal barrier coating on the blade of a gas turbine engine and calibrated the thermal stability of two layers of gadolinium zirconate-yttrium stabilized zirconia with different thicknesses using burner tests [31]. Ibrahim A. Alnaser et al. studied the stability of APS- and HVOF-coated Ni-based superalloys under different test conditions. It was concluded that the HVOF-deposited Ni-based superalloy had higher hardness and bonding strength than the APS coating. The Ni-based superalloys deposited using APS in a molten salt environment showed good corrosion resistance but poor oxidation resistance. Compared with the APS-coated superalloys, the HVOF-coated nickel superalloys showed better corrosion resistance and lower mass gain in both environments, and thus, they had better high-temperature corrosion resistance [32]. Anna Jasik’s research group thoroughly analyzed the degradation process of thermal barrier coatings containing ceramic layers with a thickness of 300 μm [33].

ODS oxide particles, especially nano-Y_2_O_3_ can be used as grain boundary pinning points to hinder the growth and crystallization of grains to effectively inhibit the growth of grains and refine grains. In addition, nano-Y_2_O_3_ can also reduce the thermal stress of Ni-based alloys by absorbing and dispersing heat, and improve its thermal stability and heat resistance. Therefore, adding nano-Y_2_O_3_ to the Ni-based alloy can significantly improve its microstructure and properties.

In addition, another method to effectively improve the wear resistance of the metal surface is to add hard particles, which can indeed further improve the tribological properties of a Ni-based metal composite coating (MCC) so that nickel-based MCC has good corrosion resistance and small wear loss rate, but it is likely to sacrifice the workpiece surface contact fatigue life and high-temperature creep resistance. The addition of 2-micron-diameter WC ceramic particles in the Ni-based alloy can play a role in refining the grain while improving the hardness and wear resistance of the alloy. This is because the WC particles themselves and the substances generated by the high-temperature reaction of the Ni-based alloy with WC particles can be used as grain boundary pinning points, hindering the growth and crystallization of grains, thus effectively inhibiting the growth of grains and refining grains. In addition, the hardness and wear resistance of WC particles are pretty high, which can increase the hardness and wear resistance of the alloy, improving its fatigue resistance and corrosion resistance.

In view of the above, an attempt in this study was made to systematically study the 2 μm-WC addition and nano-Y_2_O_3_-affected laser cladding Ni-based alloy layers. The addition of nano-Y_2_O_3_ was not only undertaken to provide the oxide dispersion strengthening effect but to provide another equally important function: to inhibit the dissolution of 2 μm scale tungsten carbide particles in the high-temperature laser melting pool. In particular, the contact fatigue and wear resistance behaviors of Ni-based alloy coatings were assessed.

## 2. Materials and Methods

Super-high-strength steel 30CrMnSiNi2A was used in this research as the substrate. This type of ultra-high-strength steel is mainly used in aerospace aircraft landing gear, where the service conditions are harsh and the steel often needs to withstand transient high-strength impact loads and linear reciprocating friction and wear, as well as having reliable salt spray corrosion resistance when undertaking coastal city navigation tasks.

Additive level: 20 wt.% μm-WC and 1.2 wt.% nano-Y_2_O_3_.

Description of various types of powder for the MMC synthesis: the morphologies and chemical compositions of commercially available Ni-based alloy and 2 μm-WC particles are shown in Figure 1 and Table 1, respectively. The purity and average particle size of the Y_2_O_3_ used in this study were 99.9% and 40 nm.

The manufacturing process of the MMCs is described as follows: the mixed powder bed of Ni-based alloy, 2 μm-WC particles, and nano-Y_2_O_3_ were cladded using a CO_2_ laser (CP4000, Convergent) with shielding nitrogen gas. The laser processing parameters are shown in Table 2. The preparation parameters of the composite coating with other components, such as adding 80 μm-WC particles and 2 μm-WC particles using a laser process, were the same as in Table 2.

The laser cladding samples were cut, ground, and polished with a Struers Tegrapol-25 (Struers, Denmark) and then characterized using field emission scanning electron microscopy (FSEM, Quanta 600FEG, FEI).

The reciprocating dry-sliding wear test was carried out on the wear instrument (UMT, center). The friction and wear volume loss rate of MMC-coated samples was calculated by measuring the optical 3D wear surface with a Uscan Custom (Nanofocus). The wear surface was immersed in acetone and the ultrasonic cleaning machine removed the friction debris.

## 3. Discussion

### 3.1. Microstructure of the As-Received Coating

Figure 2 shows the SEM micrograph of coating’s cross-section; it is clear that the amount of residual μm-WC in the laser-cladded μm-WC-Y_2_O_3_ coating was more than that in the μm-WC MMCs. By identifying the WC object boundary in Figure 2 and calculating the area proportion, the volume percentage of the composite coating occupied by WC ceramic particles in Figure 2 could be accurately analyzed. The statistical results of the μm-WC percentage for μm-WC-Y_2_O_3_ and μm-WC coatings were 14.25% and 2.37%, respectively. The average particle size for the μm-WC-Y_2_O_3_ coating was 0.452 μm and for the μm-WC coating was 0.306 μm. Then, the conclusion of Y_2_O_3_ addition limited the amount of μm-WC dissolvation that could be reached. The micron-scale WC ceramic particles had a bright white polygon pattern, as shown in the red rectangular box in Figure 2a,b.

The interdendrite area of light gray is the bright white polygonal micron-WC ceramic particles and the nano rare earth oxide Y_2_O_3_ dispersed in the melt pool, as shown by the bright blue irregular curve circled in Figure 2a. The rare earth oxide Y_2_O_3_ can play the role of heterogeneous nucleation in nickel-based alloys because Y_2_O_3_ has high thermal stability and oxidation stability and can maintain its nanometer size and dispersion state at high temperatures. Therefore, when Y_2_O_3_ is added to nickel-based alloys, they can form heterogeneous nuclei in the alloy melt, that is, dispersed particles in the alloy. These rare earth oxide Y_2_O_3_ nanoparticles can be used as grain boundary pinning points to hinder the growth and crystallization of grains to effectively inhibit the growth of grains and refine grains. In addition, the rare earth oxide Y_2_O_3_ particles can also provide a dislocation locking point, absorb and disperse the dislocation, and thus, further refine the grain. Through heterogeneous nucleation, rare earth oxides can effectively improve the microstructure of nickel-based alloys, make the grains small and evenly distributed, and thus, improve the mechanical properties and heat resistance of the alloys. In addition, the rare earth oxide Y_2_O_3_ can also provide a strengthening phase of the alloy, increasing the hardness and strength of the alloy. Therefore, the heterogeneous nucleation of the rare earth oxide Y_2_O_3_ in nickel-based alloys plays an important role in improving the properties of the alloys. To some extent, nanoscale rare earth oxide particles play the role of a thickening solvent in the melt pool. Considering the high-temperature stability of Y_2_O_3_, it is equivalent to forming a fine micro-particle barrier at the contact interface of WC and metal melt pool. The above two factors work together to prevent WC ceramic particles from dissolving in the superalloy melt pool.

The nano-Y_2_O_3_-modified microstructure with a larger amount of 2 μm-WC displayed higher wear resistance, as can be seen in Figure 3.

The four wear surfaces analyzed were immersed in acetone and removed using ultrasonic cleaning machines. During the wear process, we did not take measures to periodically blow away the wear debris. In Figure 3a, the worn surface of 2 μm-WC-Y_2_O_3_ MMC displayed a narrow and small wear scar compared with Figure 3b–d. This was maybe because 2 μm-WC-Y_2_O_3_ MMCs possessed more hard phase than the 2 μm-WC MMCs in Figure 3b and the dissolved μm-WC provided a supportive network during the wear process. The big cast spherical μm-WC MMCs displayed a worse worn surface, as can be seen in Figure 3c; this was probably due to the big particles having inherent cracks inside, which were generated during the casting procedure. However, the performance of big MMC particles was still better than in the base material 30CrMnSiNi2A.

As shown in the positions marked by the red rounded rectangle and arrow in Figure 3, the depth of friction nicks in the horizontal lateral view, the width of deeper friction nicks, and the total number of deeper friction nicks directly corresponded to the total wear volume. The order of the total wear volume corresponding to the four components could be intuitively estimated from the size of the blue-green area in the rounded corner pattern.

It is obvious that the number of deep wear scratches in Figure 3c was much more than that in Figure 3a,b. It was further shown that the hardness distribution of the nickel-based alloy composite coating with 2 μm-WC reinforced particles was more uniform, and the strength and toughness were better than that of laser coating with 80 μm-WC reinforced particles alone, regardless of whether there was nano-Y_2_O_3_ additive modification. Figure 3a shows the morphology of the 2 μm-WC strengthened nickel-base alloy composite coating modified by the nanoscale Y_2_O_3_ additive. Compared with the wear surface of the other three comparison objects, Figure 3a shows shallower scratches and less wear volume. This indicates that the nano-grade Y_2_O_3_ additive improved the 2 μm-WC-strengthened nickel-based alloy composite coating to provide the best resistance to friction and wear among the four test samples.

Contact fatigue is one of the crucial aspects in wear surface analysis; therefore, we undertook coating and base metal von Mises stress 3D simulations to provide a systematic MMC performance assessment.

### 3.2. Von Mises Stresses of Different Coatings

Regarding the modeling and simulation, the effects of the von Mises stress distribution on the WC with different shapes and sizes were further analyzed by using a half-space modeling method and a dimensionless research method. It was concluded that the laser cladding layer with the 2 μm-WC addition had a more uniform von Mises stress distribution than the spherical WC, with an average diameter of 80 μm; therefore, the laser cladding layer with the 2 μm-WC addition had better resistance to micro-cracks and transient heavy load impacts.

Figure 4 shows the von Mises stress simulation results of the 2 μm-WC-Y_2_O_3_/Ni and 80 μm-WC/Ni samples established with physical property data listed in Table 3. As shown in Figure 4a, the sample with 2 μm-WC and Y_2_O_3_ addition coating had a uniform and homogeneous stress distribution. Such results explained the mechanism of the proper WC particle size, such as 2 μm, and the amount of Ni-based alloy that could bring both excellent hard phase reinforcement for wear resistance increment and uniform von Mises stress for reliable contact fatigue performance.

In Figure 4b,c, the 3D simulation modeled ball-shape 80 μm-WC units with different diameters. Since the average diameter value was pretty small, a dimensionless 3D scale was necessary.

The von Mises stress simulation assumptions: (i)The von Mises criterion was the distorted energy theory;(ii)In order to dimensionalize the 3D space length unit, the length unit parameter a_0_ was matched, and its value was 10 times that of the maximum diameter of the corresponding WC particle;(iii)The physical property parameters of the matrix metal were not simple nickel-based alloys but needed to consider 20 wt.% μm-WC and 1.2 wt.% nano-Y_2_O_3_;(iv)Because the high temperature of the molten pool could reach 2860~2900 °C, the WC particles in the laser cladding layer were variable in size, that is, due to the burning loss, their volumes were reduced to different degrees.

In Figure 4b,c, the area encircled by the red oval was the stress concentration caused by 80 μm-WC. It can be seen that the high-stress concentration in the XOZ plane made the 80 μm-WC composite nickel-based alloy coating unsuitable for bearing impact loads. At the same time, the high-stress concentration in the XOY plane indicates that the 80 μm-WC composite nickel-based alloy coating had a weak shear-stress-bearing capacity.

Figure 5 shows the friction coefficient curve obtained from the friction and wear experiments of four components, and the detailed parameters of the friction and wear experiments are shown in Table 4. As can be seen in Figure 5, from 0 s to nearly 3800 s, the friction and wear specimen and the matching grinding ball of the pin disc structure clip were in an unstable wear surface mismatch and running-in transition process. The initial stage of this instability running process from 0 s to 1000 s was a very unstable period of friction coefficient COF, which transitioned into a relatively stable period after 1000 s and a stable COF stage until 3800 s after the end. The effective COF value we actually read was exactly the stable mean after 3800 s. As can be seen from Figure 5, the COF value of the friction coefficient of 2 μm-WC-Y_2_O_3_/Ni consistently had the minimum value of the COF value of the four specimens during the period from 800 s to nearly 7200 s. The COF of the matrix ultra-high-strength alloy steel 30CrMnSiNi2A reached the highest value of the four COF curves only after 3800 s of the steady-state threshold point, while the corresponding COF curves of the other two components also stabilized in their intervals after entering 3800 s. In short, the values of the four COF curves were basically stable after 3800 s, and the COF values at any time increased from small to large, where the corresponding component names were 2 μm-WC-Y_2_O_3_/Ni, 2 μm-WC/Ni, 80 μm-WC/Ni, and 30CrMnSiNi2A.

According to the 3D wear surface abrasion analysis in Figure 3, the frictional volume loss rate under a unit load on a unit wear trajectory was calculated, as shown in Figure 6.

As can be seen from Figure 6, the wear rate of 2 μm-WC-Y_2_O_3_/Ni was 1.3, which was significantly lower than the other three, while the matrix 30CrMnSiNi2A even reached 10 times the wear rate of 2 μm-WC-Y_2_O_3_/Ni, ranking the highest among the four components. In the remaining two components, the wear rate of 2 μm-WC/Ni was 2.4, which was better than the tribological volume loss rate of 3.5 of 80 μm-WC/Ni.

The scanning plane of the SEM test was the upper surface of the composite cladding layer perpendicular to the direction of the forming laser beam. After comparing with the addition of La [34] and Ce [35], which are similar components of rare earth oxidation particles, Y had the best thinning effect among the three. In Figure 7, the test sites marked with lake blue numbers from 027 to 033 all underwent in-depth analysis and explanation of the excellent friction and wear resistance of 2 μm-WC-Y_2_O_3_/Ni from the perspective of phase composition on the basis of atomic proportions.

The different atomic proportions at each position are shown in Table 5.

In order to demonstrate the reliability of the data in Table 5, the corresponding measured results are shown in Figure 8.

Considering Figure 2, Figure 7, Table 5, Figure 8, and the literature describing my previous research [34,35], the dark-colored dendrites marked with labels 028, 029, 030, and 032 in Figure 7 were composed of γ-Ni(Fe), while the light-colored parts between dendrites were undissolved residual 2 μm-WC, M_23_C_6_, M_7_C_3_, and Cr_2_Ni_3_. The M here refers to the metal element. For example, at the composition measurement point 033, the atomic ratio Cr:Ni = 4.51:6.34 ≈ 2:3; it can be concluded that the composition contained Cr_2_Ni_3_.

Considering that the peak temperature recorded by the infrared camera in the laser melt pool in this study could reach 2860 °C~2900 °C, while the melting point of Y_2_O_3_ is only 2439 °C, the O in the molten state of Y_2_O_3_ could react with the WC and C in the molten Ni-based alloy to form CO or CO_2_ gasification, which escaped as burn loss. Intermetallic compounds or solid solution strengthening were formed by free Y in the metal laser molten pool. However, since Y_2_O_3_ was a trace addition of 1.2 wt.% and only partially decomposed, its newly generated solid solution and intermetallic compounds were difficult to detect, even with XRD.

The reason for the high proportion of W atoms at component sampling points 027, 031, and 033 is explained here:

WC degenerated into W_2_C and metal W in the molten pool. The WC melting point is 2870 °C, which is within the peak temperature of the molten pool range of 2860–2900 °C. This explains why large amounts of undissolved and partially dissolved 2 μm-WC particles remained in the melt pool.

## 4. Conclusions

This study demonstrated that the addition of 2 μm-WC can provide better CF resistance and impact load resistance than 80 μm-WC, which means that the probability of micro-crack initiation at the interface between the hard particle WC and the alloy inside the coating was smaller.

The addition of Y_2_O_3_ to the Ni-based laser cladding layer could effectively limit the dissolution of 2 μm-WC. In addition, Y_2_O_3_ could also improve the hardness of the cladding layer through the oxide-dispersion-strengthening effect, making it more resistant to wear and corrosion, thereby improving the durability and life of the cladding layer.As a hard particle additive, the wear surface of 2 μm-WC coated with 2 μm-WC-Y_2_O_3_ had the best performance compared with the surface of the large-sized cast spherical 80 μm-WC and 30CrMnSiNi2A wear specimens. As shown in the half-space dimensionless simulation results, the von Mises stress distribution in the metal laser composite coating was very uniform and the laser coating prepared with the 2 μm-WC particle additive had better resistance to a normal impact load and tangential load than that with the 80 μm-WC additive. The heterogeneous nucleation and oxide dispersion strengthening of Y_2_O_3_ provided grain boundary pinning and dislocation stacking effects to improve the overall performance of the composite laser coating.The von Mises stress of the 2 μm-WC-Y_2_O_3_ laser coating was more uniform, while the inherent initial cracks and dense stress concentration points in the 80 μm-WC laser coating from the spherical casting process could easily cause dislocation accumulation, as demonstrated quantitatively and qualitatively, and thus, micro-crack nucleation. After the stable state of COF was reached, the order of the values of the four COF curves was stable from small to large as follows: 2 μm-WC-Y_2_O_3_/Ni, 2 μm-WC/Ni, 80 μm-WC/Ni, and 30CrMnSiNi2A.The wear rate of 2 μm-WC-Y_2_O_3_/Ni was 1.3, which was significantly lower than the frictional volume loss rate of the other three components: 2.4, 3.5, and 13.

## Figures and Tables

**Figure 1 materials-17-00121-f001:**
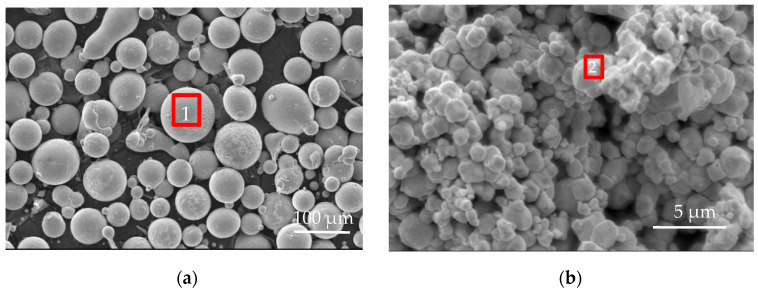
SEM images of (**a**) Ni-based alloy and (**b**) 2 μm-WC.

**Figure 2 materials-17-00121-f002:**
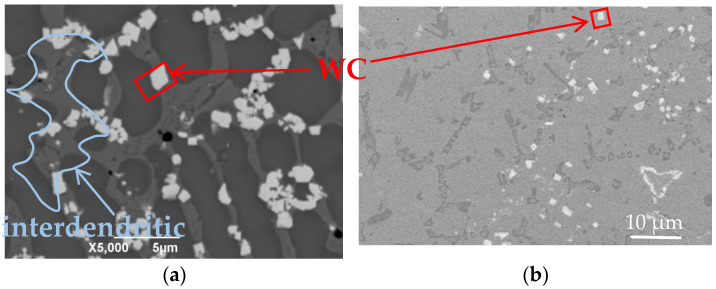
SEM images of (**a**) 2 μm-WC-Y_2_O_3_ and (**b**) 2 μm-WC MMCs without Y_2_O_3_ addition.

**Figure 3 materials-17-00121-f003:**
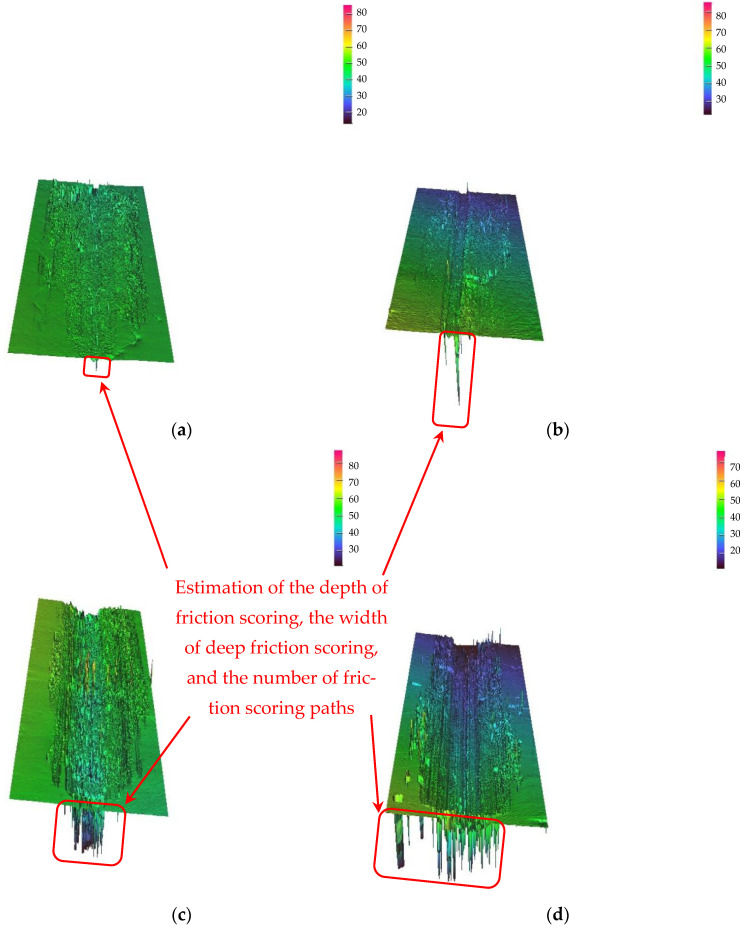
Worn surfaces of (**a**) 2 μm-WC-Y_2_O_3_, (**b**) 2 μm-WC MMCs without Y_2_O_3_ addition, (**c**) traditional 80 μm-WC addition MMCs, and (**d**) base material 30CrMnSiNi2A.

**Figure 4 materials-17-00121-f004:**
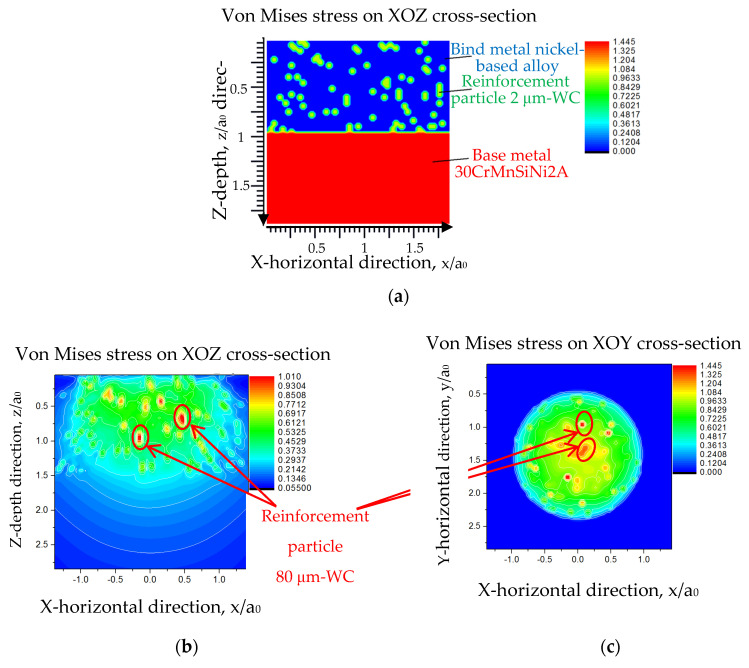
Half-space modeling and dimensionless simulation of (**a**) 2 μm-WC-Y_2_O_3_, (**b**) XOZ of traditional 80 μm-WC addition MMCs, and (**c**) XOY of traditional 80 μm-WC addition MMCs.

**Figure 5 materials-17-00121-f005:**
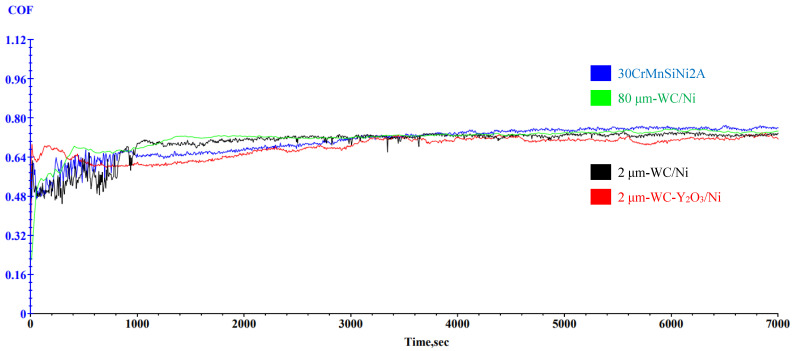
Comparison of surface friction coefficient curves of four samples.

**Figure 6 materials-17-00121-f006:**
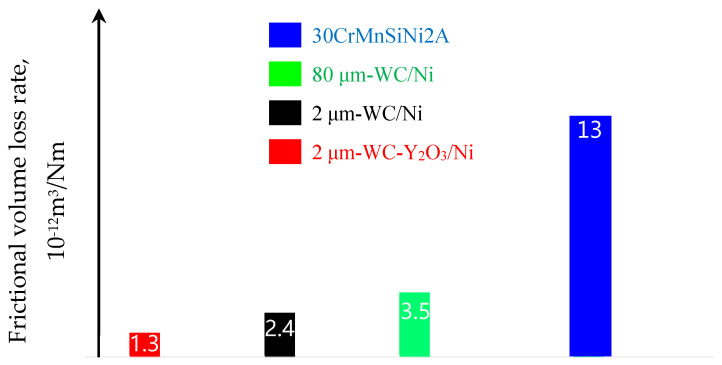
Comparison of surface friction coefficient curves of four samples.

**Figure 7 materials-17-00121-f007:**
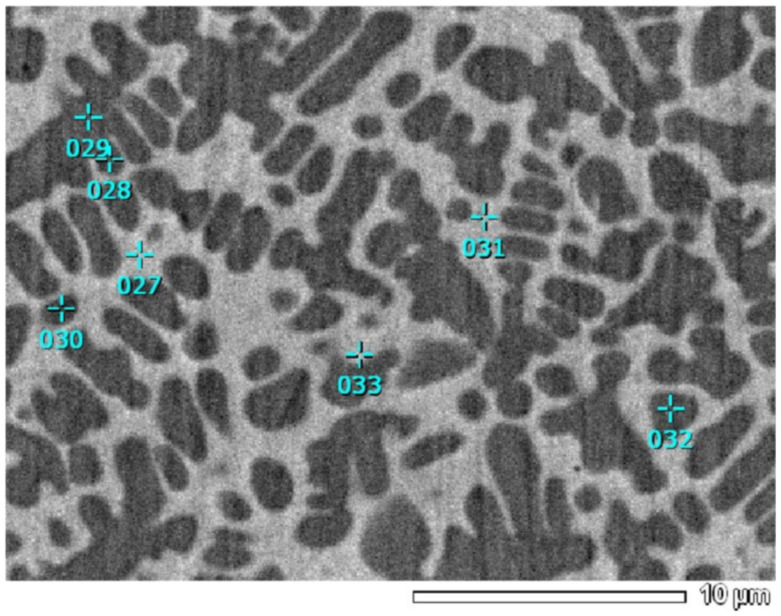
EDS phase composition analysis at multiple test points on a 2 μm-WC-Y_2_O_3_/Ni sample.

**Figure 8 materials-17-00121-f008:**
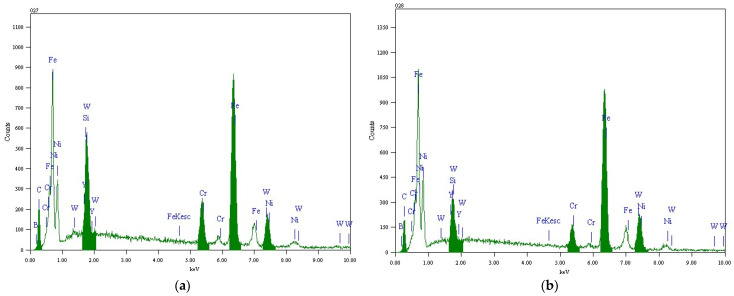
EDS results of (**a**) 027, (**b**) 028, (**c**) 029, (**d**) 030, (**e**) 031, (**f**) 032, and (**g**) 033.

**Table 1 materials-17-00121-t001:** Chemical compositions of the EDS result in Figure 1 (wt.%).

Material	Marked 1 (Ni-Based Alloy)	Marked 2 (2 μm-WC)
Ni	71.14	-
Fe	9.60	-
C	0.1	4.96
Si	4.0	-
W	-	95.04
Cr	15.15	-

**Table 2 materials-17-00121-t002:** Parameters for laser cladding [34].

Process Parameters	Operation Range
Laser type	CO_2_
Laser power (kW)	1.5–2
Laser beam diameter (mm)	~2
Melt Pool diameter (mm)	~3
Traverse speed (mm/s)	8
Overlap rate (%)	25

**Table 3 materials-17-00121-t003:** Parameters for 3D modeling.

Composite	Elastic Modulus	Poisson Ratio
μm-WC (both 2 and 80 μm)	700 GPa	0.240
Ni-based alloy	200 GPa	0.300
30CrMnSiNi2A	204 GPa	0.285
μm-WC (both 2 and 80 μm)	700 GPa	0.240
Ni-based alloy	200 GPa	0.300
30CrMnSiNi2A	204 GPa	0.285

**Table 4 materials-17-00121-t004:** Parameters for friction wear test.

Process Parameters	Operation Range
The normal load applied (N)	25
Whether measures were taken to periodically blow away the wear debris during the wear performance test	Never blew away the wear debris
Whether wear debris was removed before 3D scanning of wear marks after wear performance testing	Used acetone ultrasound to remove abrasion debris before 3D scanning of abrasion marks
Duration of a single friction and wear test on a single sample (h)	2
Opposite friction wear ball diameter (mm)	5.6
Opposite friction wear ball metal type	GCr15
Friction wear specimen contact point trajectory movement mode	Linear reciprocating mode, pin wear performance test
Opposite friction wear ball single step displacement (mm)	5
The number of repeated tests for each component of the parallel test sample	3
Lubrication condition	Dry friction
Opposite friction wear ball moving speed (mm/s)	8

**Table 5 materials-17-00121-t005:** Atomic proportions at each position of Figure 7.

Location Code 027~033	Atom	Mass%	Error%	Atom%	keV
027	B	6.11	0.17	19.09	0.183
	C	12.61	0.28	35.46	0.277
	Si	4.61	0.35	5.55	1.739
	Cr	6.47	1.12	4.20	5.411
	Fe	42.99	1.60	26.00	6.398
	Ni	11.12	2.87	6.00	7.471
	Y	1.82	1.92	0.69	1.922
	W	14.26	1.53	2.62	1.774
028	B	5.95	0.25	17.70	0.183
	C	13.06	0.35	34.97	0.277
	Si	3.16	0.49	3.62	1.739
	Cr	4.83	1.40	2.99	5.411
	Fe	54.69	2.09	31.48	6.398
	Ni	16.20	3.81	8.87	7.471
	Y	-	-	-	-
	W	2.10	1.83	0.37	1.774
029	B	2.12	0.26	7.34	0.183
	C	10.44	0.32	32.62	0.277
	Si	4.32	0.49	5.77	1.739
	Cr	5.99	1.40	4.32	5.411
	Fe	56.63	2.08	38.04	6.398
	Ni	17.78	3.81	11.36	7.471
	Y	-	-	-	-
	W	2.72	1.85	0.55	1.774
030	B	6.09	0.27	17.80	0.183
	C	13.03	0.38	34.28	0.277
	Si	4.35	0.47	4.89	1.739
	Cr	4.41	1.43	2.68	5.411
	Fe	55.82	2.16	31.58	6.398
	Ni	16.30	3.96	8.77	7.471
	Y	-	-	-	-
	W	-	-	-	-
031	B	4.57	0.18	14.66	0.183
	C	13.05	0.28	37.70	0.277
	O	0.54	0.36	1.18	0.525
	Si	4.85	0.36	5.99	1.739
	Cr	7.53	1.17	5.02	5.411
	Fe	41.04	1.67	25.49	6.398
	Ni	9.63	2.98	5.69	7.471
	Y	3.59	2.00	1.40	1.922
	W	15.20	1.58	2.87	1.774
032	B	5.53	0.27	16.42	0.183
	C	13.26	0.38	35.46	0.277
	O	0.07	0.43	0.14	0.525
	Si	3.64	0.53	4.16	1.739
	Cr	5.12	1.51	3.16	5.411
	Fe	54.00	2.26	31.05	6.398
	Ni	17.20	4.14	9.41	7.471
	Y	-	-	-	-
	W	1.18	1.98	0.21	1.774
033	B	4.65	0.18	14.79	0.183
	C	13.28	0.28	37.97	0.277
	O	0.19	0.36	0.41	0.525
	Si	5.08	0.36	6.22	1.739
	Cr	6.83	1.16	4.51	5.411
	Fe	42.64	1.66	26.23	6.398
	Ni	10.84	2.97	6.34	7.471
	Y	2.25	1.99	0.87	1.922
	W	14.24	1.59	2.66	1.774

## Data Availability

Data are contained within the article.

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
