# Peer review of "Stress Distribution in Wear Analysis of Nano-Y2O3 Dispersion Strengthened Ni-Based μm-WC Composite Material Laser Coating"

_materials, 2023, doi:10.3390/ma17010121_

Round 1
Reviewer 1 Report
Comments and Suggestions for Authors
This study investigates stress distribution and wear analysis in a nano-Y2O3 dispersion strengthened Ni-based μm-WC (micro-sized tungsten carbide) composite material laser coating. The primary aim is to assess wear resistance and contact fatigue under high load conditions for workpiece surfaces coated with various compositions: Ni-based alloy combined with nano-Y2O3, μm-WC, and different sizes of WC particles.
This research brings novelty and significance in multiple ways:
Material Composition: It explores the impact of adding nano-Y2O3 and different μm-WC particle sizes to a Ni-based alloy, particularly in laser-cladded coatings.
Evaluation of Wear Resistance: The study delves into how these added particles affect wear resistance, contact fatigue, stress distribution, and the formation of micro-cracks within the composite coating.
Enhancement of Coating Durability: It investigates how particle sizes and the inclusion of Y2O3 affect the coating's durability, emphasizing improvements in wear resistance, corrosion resilience, and overall durability.
Analysis of Stress Patterns: Using simulation and stress analysis, the research compares the performance of various compositions, particularly focusing on their impact on micro-crack initiation and wear characteristics.
This research fills a specific gap in understanding how particle size variations and nano-additions influence wear and stress behavior in these composite coatings. By observing reduced wear rates and enhanced stress distribution with the integration of nano-Y2O3 and smaller μm-WC particles, the study underscores the potential for improving mechanical properties and performance in these coatings, especially for applications under heavy loads.
Overall, this study provides detailed insights into the intricate effects of particle size and nano-additions within a unique composite material used in laser coatings. It offers valuable information on enhancing wear resistance, stress distribution, and durability, contributing significantly to the field of material science and surface engineering.
The references are relevant and numerous.
On Figure 4 (a) part of the label of the vertical scale is missing. Please correct this.
On Figure 5 the legend text is good quality (added as vectorgraphic), but the base plot is too bad quality for publication. Please change this figure, because in its current form it cannot be published.
Author Response
Figure 5. The picture has been changed to HD version and the title is marked with dark pink as shown by serial number 254 in the left row.
The scale in the Z-axis direction and the X-direction in Figure 4 (a) has been completed, and the title is marked in dark pink, as shown by serial number 228 in the left row.
Reviewer 2 Report
Comments and Suggestions for Authors
The paper ”Stress Distribution in Wear Analysis of Nano-Y2O3 Dispersion Strengthened Ni-based μm-WC Composite Material Laser Coating” presents very important aspects in terms of coatings and mechanical properties. The paper can be published in the Materials Journal after some minor corrections:
1. The abstract should be revised by summarizing the results.
2. The introduction should be updated with comparisons with other types of coatings, like APS, cold spray, or others. Suggested reference:10.3390/ma15228116
3. Figure 3 is not very well explained and is not very clear.
4. SEM and EDS analysis should be completed with XRD analysis for phase tests.
The rest is fine.
Author Response
1. Four summary results at the end have been added to the summary section, marked in dark pink.
2. Added other types of coating APS, cold spraying and other comparative studies. The reference number 10.3390/MA15228116 is added, and the reference materials of periodicals are mainly added. The added reference numbers are: 29~35, marked in dark pink.
3. A new paragraph explanation(marked in dark pink) has been added to the bottom of Figure 3, and a red text reading note has been added to the center of the figure to help understanding. However, the basic image could not be adjusted to a higher definition due to the limitations of the generation software, so we did our best to replace the Arabic numerals at the digital ruler with vector images.
4. Although XRD will solidify the results, the existing SEM and EDS data are sufficient to support the above views. Due to the seriousness of scientific research, I firmly believe that the reviewer's view is correct, but our research conditions are really very difficult. Due to the limitations of objective conditions, EDS analysis was used instead of XRD analysis, and combined with the phase composition analysis of the previous paper data, the measured temperature field data, and the melting point parameters of each component, etc., the composition analysis was jointly deduced and completed.

Reviewer 3 Report
Comments and Suggestions for Authors
It is a good paper and as such I recommend it for publication after necessary additions and corrections.

Author Response
- During the 3D simulation, we added the "Von-Mises stress himulation hypothesis"with an explanation and several description of distribution assumptions. Also we have uniformed the criteria name in this
- In the second column of Table 1 we have changed the Cr value from "15" to "15.15" (this was a typing error). Figure 5 has been replaced with a high definition version.
- On the page 7 (lines 200 - 222) Authors wrote:’ In Fig.4 (b,c), the 3D simulation modulus set ball-shape 80μm-WC unit with equal diameter. Since the average diameter value is pretty small, diameter normalization is necessary.’ What exactly is this normalization? Please explain.
The modification is: "In Fig.4 (b,c), the 3D simulation modulus set ball-shape 80μm-WC unit with different diameter. Since the average diameter value is pretty small, 3D scale dimensionless is necessary."
- Table 4 contains strange and confused descriptions of process parameters for example it is better to use 'The normal load applied [N]' instead of …/N’or ‘Opposite friction wear ball moving speed/mm/s’ (?)
The format has been changed to space + square brackets
- Figure 6 (Comparison of surface friction coefficient curves of four samples) represents rather bars than curves (?)
Figure 6 wants to state "Frictional volume loss rate" instead of COF. This object is the ratio of the final volume loss (constant value) to the load, total distance.
Editorial corrections required:
page 2: line 66 - 'didymium' and line 86 - 'will affect' better than 'will have an effect on'
Changed to "Nd and Pr" and "will affect" as suggested.
page 6: lines 208 – 212: ‘In the modeling and simulation part, we further discussed that the designed coating with 2μm-WC addition had more uniform Mises stress distribution than the conventional industrial WC with an average diameter of 80μm by means of half-space modeling method and dimensionless research methods, so the microcrack resistance and transient heavy load impact resistance would be better.’ This sentence is too long and thus incomprehensible. It seems necessary to reformulate and simplify it.
Changed to “In the part of modeling and simulation, the effects of Von-Mises stress distribution on WC with different shapes and sizes are further discussed by using half space modeling method and dimensionless research method. It is concluded that the laser cladding layer with 2μm WC addition has more uniform Von-Mises stress distribution than spherical WC with an average diameter of 80μm, so the laser cladding layer with 2μm WC addition can have better resistance to micro cracks and transient heavy load impact.” as suggested.
page 7: line 232 – ‘from 0 seconds to nearly 3800’ better than ‘from 0 seconds to close to 3800’
Changed to“nearly” as suggested.
page 8: line 244 – ‘in their own intervals
“Own”has been deleted as suggested.
